# Fewer Cardiopulmonary Complications and Shorter Length of Stay in Anterolateral Thoracolumbar Spine Exposures Using a Small-Incision Specialized Retractor System [note 1]

**DOI:** 10.3390/jcm9103119

**Published:** 2020-09-27

**Authors:** Sophia Khan-Makoid, Bruce L. Tjaden, Samuel S. Leake, Ross G. McFall, Charles C. Miller, Harleen K. Sandhu, Karl Schmitt, Kristofer M. Charlton-Ouw

**Affiliations:** 1Department of Cardiothoracic and Vascular Surgery, McGovern Medical School at The University of Texas Health Science Center at Houston (UTHealth), Houston, TX 77030, USA; sophia.khan@uth.tmc.edu (S.K.-M.); Samuel.S.Leake@uth.tmc.edu (S.S.L.); Charles.C.Miller@uth.tmc.edu (C.C.M.III); Harleen.K.Sandhu@uth.tmc.edu (H.K.S.); 2Division of Vascular and Endovascular Surgery, Cooper University HealthCare, Camden, NJ 08103, USA; Tjaden-Bruce@cooperhealth.com; 3Department of Cardiovascular Surgery, Houston Methodist Hospital, Houston, TX 77030, USA; rgmcfall@houstonmethodist.org; 4Department of Neurosurgery, McGovern Medical School at The University of Texas Health Science Center at Houston (UTHealth), Houston, TX 77030, USA; Karl.Schmitt@uth.tmc.edu; 5Memorial Hermann Hospital—Texas Medical Center, Houston, TX 77030, USA; 6HCA Houston Healthcare, Gulf Coast Division, Houston, TX 77004, USA; 7Department of Clinical Sciences, University of Houston College of Medicine, Houston, TX 77204, USA

**Keywords:** anterolateral spine exposure, XLIF, minimally invasive spine retractor

## Abstract

**Objectives:** One of the challenges of spine surgery is the need for adequate exposure of the anterolateral spinal column. Improved retractor systems with integrated lighting minimize the need for large thoracotomy, flank, or abdominal incisions. In 2013, we began using the NuVasive MaXcess^®^ system via a minimal-access lateral incision for thoracic and thoracolumbar spine exposures. These small-access approaches may not offer adequate exposure when bleeding and other complications arise. We sought to determine the feasibility and outcomes of a minimal-access retractor during anterolateral spine exposures. **Methods:** An institutional-review-board-approved retrospective chart review was performed of all patients who underwent anterolateral thoracic and lumbosacral spine exposure at an academic hospital between December 1999 and April 2017. Cervical and posterior spine exposures were not included. Information regarding patient demographics, comorbid conditions, operative techniques, exposure, estimated blood loss, length of stay, and intraoperative and postoperative complications was collected. Data for standard exposure vs. minimally invasive exposures were compared. **Results:** Between December 1999 and April 2017, 223 anterolateral spinal exposures were performed at our institution. Of those, 122 (54.7%) patients had true lateral exposures, with 22 (18%) using the minimally invasive retractor. The mean age of our patient population was 57 years (19–89), with 65 (53%) men and a mean body mass index of 29.0 (17.4–58.6). In the standard exposure group, complications occurred in 22 (22%) patients, whereas only two (9%) complications occurred in the minimal-access group. There were no significant differences in overall intraoperative and postoperative complications, except for cardiopulmonary complications, which were reduced in the minimally invasive group (*p* < 0.019). Patients with minimally invasive exposure had a significantly shorter length of stay than those with standard exposure (7 vs. 13 days, *p* = 0.001). **Conclusions:** Minimal-access techniques using advanced retractor systems are both feasible and safe compared to standard techniques allowing for similar lateral spine exposure, but with smaller incisions, fewer cardiopulmonary complications, and shorter lengths of stay.

## 1. Introduction

One of the challenges of spine surgery is the need for adequate exposure of the anterolateral spinal column. The difficulty with access to this area is its precarious location, surrounded by important neurovascular, abdominal, and thoracic structures. Access to the lumbar spine is traditionally accomplished with anterior transperitoneal or extraperitoneal approaches, and access to the thoracic spine requires intrapleural, extrapleural, or transdiaphragmatic thoracotomies. These exposures require large incisions, extensive retraction of soft tissue, solid organs, and major vascular structures, and are accompanied by various complications [1,2].

As the field of spinal surgery has progressed, minimally invasive approaches were introduced to limit the morbidity associated with these surgeries. A direct lateral approach to the spinal column allowed for equivalent access and fewer potential complications associated with traditional approaches. These exposures were further advanced with specialized retractors that allowed these approaches to be performed with incisions no larger than 3–4 cm [3].

In 2003, NuVasive introduced the MaXcess^®^ system (NuVasive, San Diego, CA 92121, USA), a three-blade system designed to allow maximum access with minimal incision. This improved retractor system with integrated lighting minimized the need for large thoracotomy, flank, and abdominal incisions. One limitation of this retractor is the ability to provide adequate access when complications arise. In contrast, this retractor claims to improve visualization with increased mobility of surgical instruments with the lateral approach’s added benefit.

In 2013, we started using the NuVasive MaXcess^®^ system (Figure 1) with a small incision for thoracolumbar spine exposures previously performed with open surgical techniques. We sought to determine the feasibility and outcomes of a minimal-access retractor during spine exposures and its effect on both intraoperative and postoperative complications for all indications. We hypothesized that minimally invasive exposures using the MaXcess^®^ retractor will lead to fewer complications than surgeries using the standard exposure.

## 2. Methods

An institutional review board-approved retrospective chart review was performed of all patients who underwent lateral thoracic and lumbosacral spine exposures at an academic hospital between December 1999 and April 2017. Anterior lumbar, cervical, and posterior exposure procedures were excluded. Information regarding patient demographics, comorbid conditions, previous surgical history, indications for surgery, operative techniques, exposure, length of stay, estimated blood loss, and intraoperative and postoperative complications were collected.

Standard exposures to the thoracolumbar spine were defined as operations conducted through a traditional lateral exposure with thoracotomy, flank, or retroperitoneal incisions. Minimally invasive exposures were defined as the cases in which the NuVasive MaXcess^®^ retractor was used. Patients were prepped in a lateral decubitus position. The spinal level was marked externally using fluoroscopy. A 3 cm incision was made at this level, and the spine was approached laterally. A K wire was then placed in the disc space of interest. Once the correct location was confirmed, serial dilators were placed over the wire. The NuVasive retractor was loaded over the dilator and the blades were deployed. An integrated neuromonitoring system can be used depending on the location and risk of nerve and spinal cord injury. The retractor has an integrated lighting system that can be attached to the blades. Periosteal soft tissue was retracted or dissected. Intercostal or lumbar vessels can be ligated or retracted (Figure 1).

Indications for surgery were classified as degenerative, deformity, trauma, tumor—primary or metastatic, infectious, and other. Length of stay was measured from the date of surgery to the date of discharge. Complications were classified into spinal, vascular, cardiopulmonary, gastrointestinal, and surgical. Complications were further stratified into iatrogenic and those resulting in serious adverse events. Serious adverse events were defined as a composite outcome of perioperative morbidity resulting in either prolonged hospital course, surgical re-intervention, infection, increased intraoperative blood loss, pulmonary embolism, or mortality.

Categorical data were compared for standard exposure vs. minimally invasive exposure by contingency table methods. Continuous data were compared by an unpaired t-test or Wilcoxon rank-sum, depending on the data’s distribution. Multivariable analyses were conducted using multiple logistic regression for categorical outcomes and multiple linear regression with appropriate variable transformations for continuous dependent variables. Primary outcome measures were number and number of complications. Types of complications were composite complications involving spinal, vascular, cardiopulmonary, gastrointestinal, and wound-related. Secondary outcome measures were the length of stay, estimated blood loss, and major adverse events. All computations were performed using SAS software version 9.4 (SAS Institute, Inc., Cary, NC, USA).

## 3. Results

Between December 1999 and April 2017, 223 anterolateral spinal exposures were performed at our institution. From this total, 101 patents had anterior exposures. Of the remaining 122 patients, 100 underwent the standard exposures, and 22 (18%) underwent the minimally invasive retractor. The system was used for 7 thoracic, 5 thoracolumbar, and 10 lumbosacral exposures. The mean age for the 122 lateral exposure patients was 57 years (range 19–89 years), with 65 (53%) mean and a mean body mass index of 29.0 (17.4–58.6) (Table 1).

A total of 232 spinal levels were exposed (mean 1.9/patient). Indications for surgical intervention included degenerative (7% vs. 9%), anatomic (22% vs. 36%), trauma (12% vs. 5%), tumor (9% vs. 0%), infection (32% vs. 27%), and other (20% vs. 23%) in standard vs. minimally invasive exposures, respectively (Table 2). Body mass index (BMI) was higher in the open surgical group than in the NuVasive group (30 vs. 26, *p* < 0.003; Table 1). There were significantly more exposures involving the lumbar region in the minimal access group (46% vs. 20%, *p* < 0.012; Table 2).

There was no significant difference in estimated blood loss between the groups: 703 mL in the standard exposure group compared to 638 mL in the minimally invasive group (*p* = 0.645). Patients with minimally invasive exposure had a significantly shorter length of stay than those with standard exposure (7 vs. 13 days, *p* = 0.001, Table 3). When the patients without complications were excluded from the analysis, the minimally invasive exposure group continued to have a significantly shorter length of stay (6.8 ± 5.2 days (*n* = 20) vs. 12.2 ± 10 days (*n* = 78), *p* = 0.001). After adjustment for time effect or date of surgery, the minimally invasive group still had a significantly shorter length of stay in a generalized linear model (Table 4).

There were 22 (22%) complications in the standard open approach compared to 2 (9%) complications in the minimally invasive group (*p* = 0.240, Table 3). One mortality in the standard exposure group was due to adult respiratory distress syndrome (ARDS) in an infected spine case. (Table 5). There was no significant differences in overall intraoperative and postoperative complications, except for cardiopulmonary complications, which were lower in the minimally invasive group (*p* < 0.019). No cardiopulmonary complications, wound complications, gastrointestinal complications, or deaths were observed in the minimally invasive group. There was one dural tear and one intercostal vein injury that were both repaired intraoperatively in the minimal access group.

Although the treated BMI and anatomic levels differed between groups, these variables were not associated with complications or length of stay. In multivariable analysis, these risk factors did not affect the relationships between the minimally invasive approach and complications or lengths of stay.

## 4. Discussion

We compared the feasibility and safety of minimal access exposures using the NuVasive retractor to standard exposures. The results of this study indicated that the minimally invasive retractor is versatile enough to provide adequate exposure for various indications and procedures. There was no statistically significant difference in the number of overall complications in this patient population, suggesting that minimally invasive approaches are non-inferior to standard surgical approaches regarding complications.

When stratified to complication, based on type, there were five (5%) patients in the standard surgical group that had postoperative pulmonary embolisms and none in the minimally invasive group, which we think is clinically important. There was a significantly decreased length of stay using the minimal access retractor. Given the retrospective nature of this study, it is difficult to ascertain what factors may have contributed to the fewer pulmonary embolisms. However, one of the known risk factors for pulmonary embolisms is immobility. Therefore, it is conceptualizable that patients with minimal incisions and shorter lengths of stay were mobilized sooner than patients with larger incisions and longer lengths of stay.

Cardiopulmonary complications in this study population could be attributed to a thoracotomy procedure and not from the spine exposure. However, when exclusively considering the patients who had spine exposures at the thoracic level, there were no cardiopulmonary complications in the minimally invasive group. There was one patient with a lumbosacral exposure with a lung laceration and no thoracotomy involved.

Historical publications have suggested a sizeable risk of vascular injury during anterior spinal exposure, usually involving the iliac vessels [4]. Nourian et al. noted major venous injuries in 4.6% of cases and major arterial injuries in 1.6% [5]. When minor injuries are included, the historical rate of vascular injury exceeds 10% [6,7,8]. The use of the minimally invasive retractor allows for a lateral exposure to the spinal column, thus avoiding potential hazards of iliac vessels when performing more anterior exposures. This was one of the reasons for the enthusiasm for lateral exposures. With the lateral exposures, bleeding complications occur with intercostal and lumbar vessels. One of the major concerns regarding the use of the minimally invasive retractor is that it might provide inadequate exposure if bleeding was to occur. One intercostal vein injury in the minimally invasive group was adequately exposed and controlled with suture ligation and did not require conversion to a larger incision.

Several studies reported the use of lateral exposures for deformity and degenerative indications. Our study indicated that the minimal access retractor can be used for nearly all indications. In our study population, albeit small, the retractor provided adequate exposure for the above-stated indications and in cases of trauma, infection, and redo spinal surgery. The retractor was not used for oncologic resection. Aside from tumor resection, no surgical indications, anatomic barriers, or patient parameters that limited the retractor’s use were found. During the initial experience with the retractor, there were no conversions to open exposure.

Vascular and cardiothoracic surgeons are experienced in thoracic and retroperitoneal exposures and play a crucial role in spine exposures. Tomita et al. published a retrospective review of non-vascular operations requiring vascular surgeon assistance at a single institution, excluding trauma cases or inferior vena cava filter placement cases. Fifty-two percent of cases requiring vascular assistance were spine exposure cases [9]. It has been our experience that the minimal-access retractors are an easily mastered tool, and we experienced success in implementing it into our program.

This study was the first to compare traditional open incisions to minimal access spine exposures for a wide variety of indications. One of the limitations of this study was its retrospective nature. It was difficult to determine if any surgeon biases limited the use of the minimally invasive retractor as this was a novel technique at the beginning of this study period. The decision to use the minimally invasive approach instead of standard exposure was based solely on exposure surgeon and neurosurgeon preference. No anatomic or surgical indication constraints were a priori applied. This can lead to selection bias and is a limitation of our paper. Additionally, there was a practice pattern change during this study. The technique was first implemented in our institution in 2013, followed by a 14-year lag period where only standard exposures were performed. This is an inherent limitation to retrospective reviews where surgical practice is advanced during the study.

An additional limitation of this study is the small number of patients resulting in a lack of statistical significance. With growing interest in minimally invasive approaches and with continued success with this system, significance in the complication rates in favor of the minimally invasive retractor may become apparent as this is a continued area of research.

## 5. Conclusions

Minimal access techniques using advanced retractor systems are both feasible and safe compared to standard techniques allowing for similarly adequate exposure, but with smaller incisions, fewer cardiopulmonary complications, and shorter lengths of stay.

## 6. Article Highlights

### 6.1. Type of Research

Key Findings: There was no significant difference in overall intraoperative and postoperative complications, except for cardiopulmonary complications, which were reduced in the minimally invasive group (*p* < 0.019). Patients who had a minimally invasive exposure had a significantly shorter length of stay than those with standard exposure (7 days vs. 13 days, *p* = 0.001.)

Take-Home Message: Using minimally invasive retractor systems for anterolateral spine exposures provides adequate exposure with shorter lengths of stay, smaller incisions, and fewer cardiopulmonary complications.

### 6.2. Table of Contents Summary

This single-center retrospective chart review showed fewer cardiopulmonary complications and shorter lengths of stay with the minimally invasive approach compared to the standard open surgery for anterolateral spine exposures. The authors suggest that the use of this minimally invasive approach continues to be used and refined in the field of spine exposure.

## Figures and Tables

**Figure 1 jcm-09-03119-f001:**
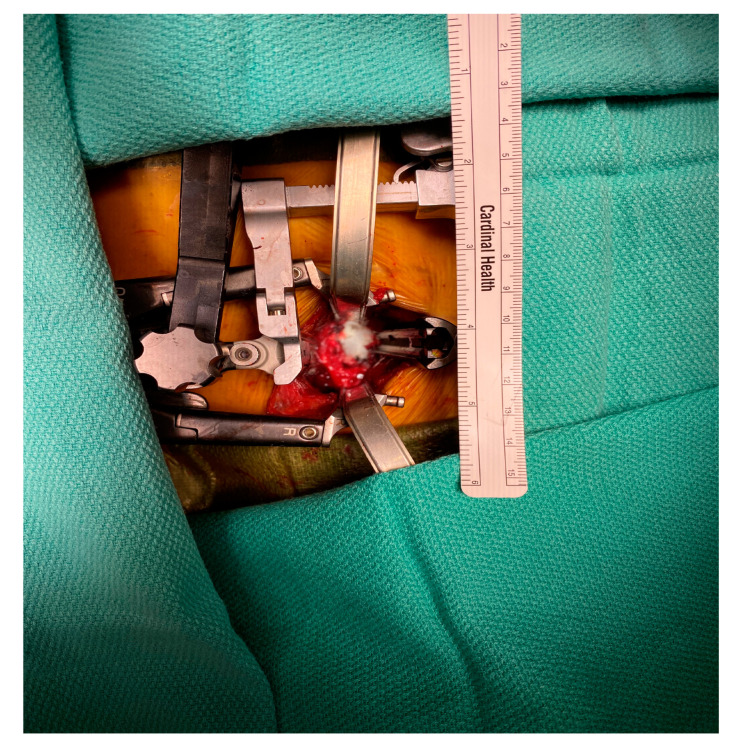
The small incision required by the minimally invasive retractor to allow for adequate exposure for lateral spine exposures.

**Table 1 jcm-09-03119-t001:** Demographic data ^†^.

Variable	Standard Lateral Exposure	Minimally Invasive Lateral Exposure	*p-*Value
	*n* = 100	*n* = 22	
Age	54.4 ± 15.1	57.6 ± 20.2	0.414
Male sex	51 (51.0)	14 (63.6)	0.282
BMI	29.8 ± 8.5	25.7 ± 4.7	0.003
Hypertension	51 (51.0)	12 (54.6)	0.763
Hyperlipidemia	23 (23.0)	6 (27.3)	0.670
Diabetes	19 (19.0)	5 (22.7)	0.768
Coronary Artery Disease	6 (6.0)	2 (9.1)	0.635
Previous Surgery *	39 (39.0)	9 (40.9)	0.874
Abdominal	25 (25.0)	7 (31.8)	0.510
Spinal	17 (17.0)	3 (13.6)	1.000
Vascular	3 (3.0)	0 (0.0)	1.000
Retroperitoneal	4 (4.0)	1 (4.6)	1.000

^†^ Values are mean + standard deviation or number of patients (%). * Included patients may have had more than one previous surgery. BMI, body mass index.

**Table 2 jcm-09-03119-t002:** Operative data ^†^.

Variable	Standard Lateral Exposure	Minimally Invasive Lateral Exposure	*p-*Value
	*n* = 100	*n* = 22	
Indication			
Degenerative	7 (7.0)	2 (9.1)	0.664
Deformity	22 (22.0)	8 (36.4)	0.157
Traumatic	12 (12.0)	1 (4.6)	0.460
Tumor	9 (9.0)	0 (0.0)	0.362
Infectious	32 (32.0)	6 (27.3)	0.665
Other	20 (20.0)	5 (22.7)	0.775
Anatomic Levels			
Thoracic	48 (48.0)	7 (31.8)	
Lumbar	20 (20.0)	10 (45.5)	0.043
Thoracolumbar	32 (32.0)	5 (22.7)	
Mean No. Levels Treated	2.9 + 0.9	2.6 + 0.7	0.078
Rib Resection	41 (41.0)	9 (40.9)	0.994
Estimated Blood Loss (mL)	702.9 + 568.9	638.2 + 674.7	0.645

^†^ Values are mean + standard deviation or number of patients (%).

**Table 3 jcm-09-03119-t003:** Outcomes ^†^.

Variable	Standard Lateral Exposure	Minimally Invasive Lateral Exposure	*p-*Value
	*n* = 100	*n* = 22	
Postoperative LOS	12.7 + 9.6	7.2 + 5.2	0.001
Any Complication	22 (22.0)	2 (9.1)	0.240
Spinal Complication	1 (1.0)	1 (4.6)	0.329
Vascular Complication	1 (1.0)	1 (4.6)	0.329
Cardiopulmonary Complication	13 (13.0)	0 (0.0)	0.019
Wound Complication	5 (5.0)	0 (0.0)	0.584
GI Complication	2 (2.0)	0 (0.0)	1.000
Iatrogenic Complication	4 (4.0)	2 (9.1)	0.295
Serious Adverse Event	14 (14.0)	1 (4.6)	0.302
Death	1 (1.0)	0 (0.0)	1.000

^†^ Values are mean + standard deviation or number of patients (%). LOS, length of stay (days); GI, gastrointestinal.

**Table 4 jcm-09-03119-t004:** Multivariable analysis for independent predictors of postoperative length of stay.

Parameter	Estimate	Standard Error	*p*-Value
Intercept	9.004914443	17.94	0.617
Minimally invasive exposure	−5.700149448	2.40	0.019
Cohort time	0.000192519	0.00	0.837

**Table 5 jcm-09-03119-t005:** Complication types in the standard open approach.

Standard Lateral Exposure	
Spinal	
	Cage misalignment (*n* = 1)
Vascular	
	IVC Injury (*n* = 1)
	Retroperitoneal Hematoma (*n* = 1)
	Retroperitoneal Lymphocele (*n* = 1)
Cardiopulmonary	
	Pulmonary Embolus (*n* = 5)
	Acute Respiratory Distress Syndrome (*n* = 1)
	Pleural Effusion (*n* = 3)
	Respiratory Failure (*n* = 1)
	Lung Laceration (*n* = 1)
	Pericardial Effusion (*n* = 1)
Wound Complication	
	Incisional Hernia (*n* = 1)
	Surgical Site Infection (*n* = 1)
	Wound Dehiscence (*n* = 1)
Gastrointestinal Complication	
	Post-operative ileus (*n* =2)

IVC, inferior vena cava.

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
