# Peer review of "Fewer Cardiopulmonary Complications and Shorter Length of Stay in Anterolateral Thoracolumbar Spine Exposures Using a Small-Incision Specialized Retractor System†"

_jcm, 2020, doi:10.3390/jcm9103119_

Round 1
Reviewer 1 Report
Thank you to the authors for the submission of the manuscript “Fewer pulmonary complications and shorter length of 2 stay in anterolateral thoracolumbar spine exposures 3 using a small-incision specialized retractor system” to the Journal of Clinical Medicine.
The study investigates the impact of a minimally invasive surgical approach vs. standard of care for antero-lateral spine surgery on postoperative complications.
Please address some of the major and minor comments listed below:
Major comments:
- Page 2, ln 71: You have stated the Aim of the study however please add an a priori hypothesis.
- Page 3, ln 103: I think it’s important to clearly define your primary outcome. "Types of complication" in ambiguous. Is the primary outcome a composite outcome of several complications (then state clearly which ones). Is pulmonary complications a composite outcome or just pulmonary embolus. If just the latter, clearly you are missing other complications like HAP/VAP. Please consider refining your primary outcome. I would also have a separate sub-heading for primary and secondary outcome before the statistical section. That way the reader can understand what your dependent variable is. I would also consider adding a sub-heading for covariates that you included in the multivariable analyses. This way dependent and independent variables are clearly defined.
- Page 4, Table 3: once again you need to define what constitutes each of the different complications.
Minor comment:
- Page 2, ln 51: comma missing between neurovascular and abdominal
- Page 3, ln 88: Include both nerve and cord injury
- Page 5, ln 147: p value is missing
Author Response
- Page 2, ln 71: You have stated the Aim of the study however please add an a priori hypothesis.
We hypothesize that minimally invasive exposure using the MaXcess retractor will lead to fewer complications than surgeries using the standard exposure. Page 2: ln:72-73
- Page 3, ln 103: I think it’s important to clearly define your primary outcome. "Types of complication" in ambiguous. Is the primary outcome a composite outcome of several complications (then state clearly which ones). Is pulmonary complications a composite outcome or just pulmonary embolus. If just the latter, clearly you are missing other complications like HAP/VAP. Please consider refining your primary outcome. I would also have a separate sub-heading for primary and secondary outcome before the statistical section. That way the reader can understand what your dependent variable is. I would also consider adding a sub-heading for covariates that you included in the multivariable analyses. This way dependent and independent variables are clearly defined.
Primary outcome measures were number and types of complications. Types of complications were composite complications involving spinal, vascular, cardiopulmonary, wound-related and gastrointestinal complications. Page 3 ln104-106
Please see new tables 4 and 5 which list specific complications. Page 8, Ln 260 -293
Table 4: Complication Type: Standard Lateral Exposure |
|
|
|
Spinal
Cage misalignment (n=1)
Vascular
IVC Injury (n=1)
Retroperitoneal Hematoma (n=1)
Retroperitoneal Lymphocele (n=1)
Cardiopulomary
Pulmonary Embolus (n=5)
Acute Respiratory Distress Syndrome (n=1)
Pleural Effusion (n=3)
Respiratory Failure (n=1)
Lung Laceration (n=1)
Pericardial Effusion (n=1)
Wound Complication
Incisional Hernia (n=1)
Surgical Site Infection (n=1)
Wound Dehiscence (n=1)
Gastrointestinal Complication
Post-operative ileus (n =2)
Table 5: Complication Type: Minimally Invasive Lateral Exposure |
|
|
|
Spinal
Dural Tear (n =1)
Vascular
Intercostal Vein Injury (n=1)
- Page 4, Table 3: once again you need to define what constitutes each of the different complications.
Please see new tables 4 and 5 which list specific complications.
Minor comment:
- Page 2, ln 51: comma missing between neurovascular and abdominal – completed
- Page 3, ln 88: Include both nerve and cord injury – completed.
- Page 5, ln 147: p value is missing - revised:
When stratified to complication based on type there were 5 (5%) patients in the standard surgical group that had postoperative pulmonary embolisms and none in the minimally invasive group, which we believed to be clinically important. Page 5, ln151-153
Reviewer 2 Report
This manuscript includes a relevant message for readers. I have a few comments that could be considered by the authors.
- In lines 93-94 I could read that “complications were classified into spinal, vascular, cardiopulmonary…” but in Table 3, there is no reference to cardiac complications and the authors present only 13 pulmonary complications. My understanding is that there were no cardias complications in the series (atrial fibrillation, ischemia, etc); if that is the case, it should be stated somewhere in the text.
- Outcomes are quite the same in both series, except for the number of pulmonary complications (13 vs nil). That is a relevant finding since series are comparable in terms of risk for adverse events. Nevertheless, there is not enough information in the text on the kinds of pulmonary adverse events in the series. I expected to find in the text the types of pulmonary complications: pneumonia, embolism, atelectasis, pleural effusion, pneumothorax, etc. That information is not irrelevant since some complications could result for prolonged preoperative hospital stay, smoking status (not included in Table I), surgical technique, etc.
- The message of the manuscript is that the new technique decreases pulmonary complications and hospital staying. To me, both could be the same: having no complications implies earlier hospital discharge. You could present separately data on hospital stay for complicated and no complicated cases and maybe also a shorter stay could be demonstrated for cases in the new technique arm.
Author Response
Reviewer 2
- In lines 93-94 I could read that “complications were classified into spinal, vascular, cardiopulmonary…” but in Table 3, there is no reference to cardiac complications and the authors present only 13 pulmonary complications. My understanding is that there were no cardias complications in the series (atrial fibrillation, ischemia, etc); if that is the case, it should be stated somewhere in the text.
There was one pericardial effusion which reflects the cardiopulmonary complication. This is further clarified in Table 4.
- Outcomes are quite the same in both series, except for the number of pulmonary complications (13 vs nil). That is a relevant finding since series are comparable in terms of risk for adverse events. Nevertheless, there is not enough information in the text on the kinds of pulmonary adverse events in the series. I expected to find in the text the types of pulmonary complications: pneumonia, embolism, atelectasis, pleural effusion, pneumothorax, etc. That information is not irrelevant since some complications could result for prolonged preoperative hospital stay, smoking status (not included in Table I), surgical technique, etc.
The new tables 4 and 5 will add further clarification and are critical to the reviewer’s comments.
- The message of the manuscript is that the new technique decreases pulmonary complications and hospital staying. To me, both could be the same: having no complications implies earlier hospital discharge. You could present separately data on hospital stay for complicated and no complicated cases and maybe also a shorter stay could be demonstrated for cases in the new technique arm.
We agree and we added an additional analysis to the manuscript:
Additionally, when those patients without complications were excluded from analysis, the minimally invasive exposure group continued to have a significantly shorter length of stay (6.8 +/- 5.2 days (n=20) vs. 12.2 +/- 10 days (n=78), p-value: 0.001) page 4, line 124-127
Round 2
Reviewer 2 Report
No additional comments.
Author Response
Thank you for the detailed review. We accepted and made the suggested edits.
